# Green and Sustainable Treatment of Urine Wastewater with a Membrane-Aerated Biofilm Reactor for Space Applications

**Chengbo Zhan** [1,2,*] , **Liangchang Zhang** [3] , **Weidang Ai** [3] and **Wenyi Dong** [1]

1    School of Civil and Environmental Engineering, Harbin Institute of Technology (Shenzhen), Shenzhen 518055, China
2    Space Institute of Southern China, Shenzhen 518117, China
3    China Astronaut Research and Training Center, Beijing 100094, China
*    Correspondence: chengbo.zhan@gmail.com

**Abstract:** Sustainability has been a concern of survival for future long-term manned space missions. Therefore, the wastewater generated by the crew members, containing urine and hygiene wastewater, should be treated with appropriate biological processes to promote recycling efficiency. In this study, we developed a membrane-aerated biofilm reactor (MABR) that could achieve up to 96% total organic carbon (TOC) removal efficiency and up to 82% denitrification efficiency for an influent with 370–390 mg/L TOC and 500–600 mg/L total nitrogen (TN) without additional carbon source or sludge discharge. The nitrogen removal rate was about 100 mg N $L^{-1}$ $d^{-1}$. Metagenomic analysis indicated the presence of a variety of nitrifying, denitrifying, and anammox bacteria in the microbial community and existence of functional genes in nitrification, denitrification, and anammox pathways.

**Keywords:** controlled ecological life support system; urine wastewater treatment; nitrogen removal; membrane-aerated biofilm reactor

## 1. Introduction

The space station has been the most expensive class of human architecture. The operation of a space station requires constant resupply of food, water, and other consumables for the crew members on shift. For future manned space missions with longer distance, such as the expedition for the Moon and Mars, sustainability would become a concern about survival. Technically speaking, a Mars expedition mission would possibly last 30 or more months [1], but the launch window for a Earth–Mars trip emerges only once per 26 months. Therefore, the recycling efficiency for all materials should be as high as possible to reduce logistic costs. To meet this requirement, the development of a controlled ecological life support system (CELSS) has been an important issue for space agencies all around the world [2]. In a CELSS with high recycling efficiency, the cycle of air, water, food, and nutrients could be maintained with minimum external supply.

To date, a completely closed CELSS that could last months to years has not been achieved yet. The ISS urine wastewater was chemically stabilized and then distilled by vapor compression distillation (VCD) process, which was restricted to the solubility limit of various compounds inside [3]. During this process, about 15% of the total volume of this stream would remain as urine brine, which could not be further concentrated by distillation [4]. $NH_4^+$-N was hard to be chemically converted under mild conditions; therefore, the stream was chemically stabilized with strong acids and oxidants to prevent the evaporation of $NH_3$ with water vapor. The brine was hazardous and could not be further distilled, leading to loss of water. Therefore, in theory, more water could be recovered if sustainable technologies with lower consumption, milder operation conditions, and no hazardous products, such as advanced oxidation processes (AOPs) and biological technologies, could substitute the current chemical pre-treatment and physical distillation treatment on urine wastewater [5,6]. In this study, we focused on biological methods as substitution.

Biological technologies could reduce the maintenance cost of water recovery. Based on equivalent system mass (ESM) analysis, the deployment of biological nitrification process could significantly decrease the ESM cost of downstream water polishing systems [7], because removing $NH_4^+$-N by ion exchange would cost more system mass. Meanwhile, the nitrogen gas generated from denitrification could somehow compensate for the atmosphere loss in the cabin. From 2008 to 2015, 856 kg of $N_2$ had been supplied to ISS (360 g/d) [8]. For each crew member, the daily amount of nitrogen excreted was about 7–16 g N [9]; if three crew members were on shift and denitrification efficiency was adequately high, about 10% of the daily loss of $N_2$ could be compensated.

The reduced gravity conditions in space would have great limitation to the mixing and separation of different phases (gas/liquid/solid). A possible solution would be the membrane-aerated biofilm reactor (MABR), which had attached biofilm on aeration membrane surface and oxygen was diffused to the biofilm from the lumen, so the gas supply process was compatible with reduced gravity conditions [10]. Moreover, the biofilm structure in MABR was different from conventional sludge granules. The oxic regions in MABR were near fiber surface and to the bulk liquid side anoxic regions were located. Thus, the oxygen and organic substrates had a counter diffusion towards the biofilm. This configuration would inhibit the overgrowth of heterotrophic bacteria, leading to the accumulation of autotrophic nitrifying bacteria at oxic regions and utilization of nitrites and nitrates as electron acceptors at anoxic regions. As a result, the efficiency of simultaneous nitrification and denitrification (SND) would be enhanced.

Initially, a tubular reactor was developed as a bioreactor that could minimize the gravity effect [11], but it was frequently blocked by biomass and thus required too much maintenance time. MABR would not have this problem, since the biofilm was attached on the outer surface [12], and several configurations had been tested, including static or rotational [13], nitrification or simultaneous nitrification/denitrification [14,15]. Currently, the research team at Texas Tech University had developed a rectangular MABR system (rCOMANDR) that adapted to the flight cargo for future experiments in space [16]. This reactor utilized nonporous silicone tubes as the growth substrate for biofilm as well as the aeration platform, where $O_2$ (from pure $O_2$ or air) diffused from the lumen side to the shell side. This system could be operated under high dissolved oxygen level for organic removal and organic nitrogen oxidation or under low dissolved oxygen level to achieve and remove part of the total nitrogen [17].

In this study, an MABR module was fabricated with microporous hydrophobic polyvinyldifluoride (PVDF) hollow fibers as the aeration units. This would lead to a reactor compatible for both flight stage and planetary base scenario. For nonporous membranes, the gas was diffused through the membrane [18], and the oxygen transfer efficiency could be easily promoted by elevating the aeration pressure, whereas for microporous membranes, the aeration pressure had to be controlled under bubble point [19]. However, this limitation of aeration pressure threshold would not be a drawback when the bulk liquid should be anoxic, in which dissolved oxygen level had to be kept low so nitrate as electron acceptors would be favored, and low aeration pressure also indicated lower energy consumption. In this study, early planetary base (EPB) wastewater used in a controlled ecological life support system experiment with four crew members [20] was treated with MABR. The EPB wastewater contained hygiene wastewater and 10 vol% of urine, and the short-term goal for this study was to achieve 50% of nitrogen removal and 90% of total organic carbon (TOC) removal. Additionally, the concentration of anionic surfactants in effluent should be lower than 5 mg/L (according to the irrigation water quality standards for farmland of National Standards of People's Republic of China, GB 5084-2021) since it could be potentially used for hydroponic systems. To our knowledge, this study was the first case utilizing PVDF microporous hollow fibers in MABR configuration and carrying out metagenomic analysis to obtain information on functional microorganisms and genes for urine wastewater in space missions. In the future, if a flight experiment of this MABR system was conducted, the comparison of metagenomic between space and ground

samples would help the understanding of space environmental effects on the microbial community for nitrogen removal.

## 2. Material and Methods

### 2.1. Materials

The reagents used in the experiment were purchased from Shanghai Macklin Biochemical Co., Ltd. with AR grade or above, including chemicals for nutrient media (listed in Table S1) and sodium bicarbonate ($NaHCO_3$). Hygiene products for waste stream preparation were purchased from JD.com, including toothpaste (Lengsuanling, Dencare), shampoo (Rejoice, P&G), facial cleanser (Mentholatum), shower gel (Safeguard, P&G), hand soap, and laundry detergent (both from Blue Moon Industrial Co. Ltd.). Urine was collected by a 25 L bucket put inside a toilet compartment of one male bathroom located inside the institute building during the project, and the collected urine was stored at 4 °C before waste stream preparation. Most of the urine consumed in this study was donated by the authors. The nutrient media used for sludge activation was prepared according to the composition listed in Table S1, for which a stock solution of concentrated micronutrients was used. The influent stream (EPB wastewater) was prepared according to Table 1. After preparation, the EPB wastewater would be stored at 4 °C and equalized for at least 12 h before fed to the influent tank. The solvent for all steps involved in nutrient media and influent stream preparation was tap water.

**Table 1.** Composition and concentration of characteristic pollutants of full strength EPB wastewater. (TOC: total organic carbon, TN: total nitrogen, COD: chemical oxygen demand, LAS: linear alkylbenzene sulfonates).

| Composition | | Concentration (mg/L) | |
|---|---|---|---|
| toothpaste | 0.1 g/L | TOC | 400–650 |
| shampoo | 0.1 g/L | TN | 500–800 |
| facial | 0.1 g/L | COD | 500–700 |
| shower | 0.1 g/L | LAS | 20–40 |
| hand soap | 0.1 g/L | | |
| laundry | 0.15 g/L | | |
| urine | 10 vol% | | |

### 2.2. Reactor Setup and Operation

A home-made MABR module was fabricated to finish the experiment. The module was made of 1440 hydrophobic PVDF hollow fibers, acrylic shell and epoxy resin sealing. The PVDF fibers were kindly provided by Shenzhen Dreamemway Environmental Products Co., Ltd. (Shenzhen, China) with 1.9 mm outer diameter, 0.5 mm thickness, and 0.1 µm nominal pore size. Both distal ends of the fibers were left open, and the gas flow had flow-through configuration. The details of the module fabrication can be found in the supplemental information (Figure S1). The length contacting the bulk liquid for the fibers was approximately 16 cm, and the overall surface area of the hollow fibers was about 1.37 m$^2$. Scheme S1 illustrates the configuration of the MABR. The reactor was a cuboid acrylic container with size of 37 cm × 37 cm × 30 cm, and the MABR module was set about the center of the tank. Additional weight stacks were attached to the bottom of the module to prevent it from floating. Two submerged pumps were put on the bottom of the reactor to provide circulation. The influent was fed to the reactor tank continuously by a peristaltic pump, and the effluent was collected in a bucket at a lower position by overflow. Since the influent stream contained urine, the influent tank was kept inside a small fridge to prevent unpleasant smell spreading and stream evaporation. The effective volume of the reactor tank was estimated to be 22 L, after the module, pumps, and probes were in position. A pressure gauge was set before the inlet to monitor the aeration pressure and the gas flow rate of inlet air and exhaust air was measured by two glass rotameters.

The seed sludge was acquired from return sludge tank on December, 2020 from Hengling Wasterwater Treatment Plant, Shenzhen, China, and the total suspended solid concentration was approximately 22,000 mg/L. About 10 L of concentrated sludge was inoculated to the reactor tank and 12 L of nutrient media was added to the reactor, and the sludge was aerated and fed with a solution of carbon source, nitrogen source, and micronutrients (Table S1). After a decrease in $NH_4^+$-N and emergence of nitrite ($NO_2^-$-N) and nitrate ($NO_3^-$-N) was observed in effluent, the reactor was ready to be fed with diluted EPB wastewater.

After sludge inoculation, the reactor was fed with nutrient media to promote the biofilm growth on the MABR module. Then, the influent was switched to low strength EPB wastewater to acclimate the sludge. The influent strength was progressively increased according to the schedule listed in Table 2. After 4 phases of acclimation (P1–P4), full strength EPB wastewater was fed into the reactor. During P5-1 to P5-3, the performance for three hydraulic retention time (HRT) was tested. For each phase, the temperature, pH, electrical conductivity (EC), and dissolved oxygen (DO) data was logged by an Orion$^{TM}$ Versa Star Pro$^{TM}$ multiparameter benchtop meter (Thermo Fisher Scientific, Waltham, MA, USA). Water samples from influent and effluent tanks were acquired 3 times per week to evaluate the pollutant removal efficiency. Water quality data was measured, including total organic carbon (TOC), total nitrogen (TN), $NH_4^+$-N, $NO_2^-$-N, $NO_3^-$-N, and linear alkylbenzene sulfonates (LAS). Sludge samples were taken on day 59 (P3), day 111 (P5-1), day 131 (P5-2), and day 165 (P5-3) for metagenetic analysis. No sludge was discharged during the experimental period.

**Table 2.** Experimental schedule for MABR. The influent strength indicated the volume fraction of EPB wastewater inside (e.g., 40% strength EPB meant the influent contained 40 vol% of EPB wastewater and 60 vol% of tap water).

| Phase | Period | Influent | HRT (d) |
|---|---|---|---|
| Phase 0 (P0) | Day 1–7 | Nutrient media | 0.92 |
| Phase 1 (P1) | Day 8–20 | 10% strength EPB | 2.24 |
| Phase 2 (P2) | Day 21–37 | 25% strength EPB | 2.24 |
| Phase 3 (P3) | Day 38–61 | 40% strength EPB | 2.24 |
| Phase 4 (P4) | Day 62–89 | 70% strength EPB | 2.99 |
| Phase 5-1 (P5-1) | Day 90–117 | full strength EPB | 2.99 |
| Phase 5-2 (P5-2) | Day 118–140 | full strength EPB | 3.94 |
| Phase 5-3 (P5-3) | Day 141–198 | full strength EPB | 4.83 |

The profiles for pH, temperature, DO, and EC of the MABR system through all phases are shown in Figure 1. Throughout the experiment, the aeration pressure was kept around 5–6 kPa, and the aeration flow rate of air was about 1500 mL/min at inlet and 1100 mL/min at outlet. In general, the pH stayed within the range of 6–8, and the temperature was within the range of 23–30 °C. Due to a travel plan, the influent was ceased during day 65–72, and a boost of DO and a drop of pH was observed, which could be attributed to the exhaustion of pollutant in the reactor. After the influent resumed, the pH and DO gradually turned back. The DO exhibited a declining trend as the load of substrates concentration increased.

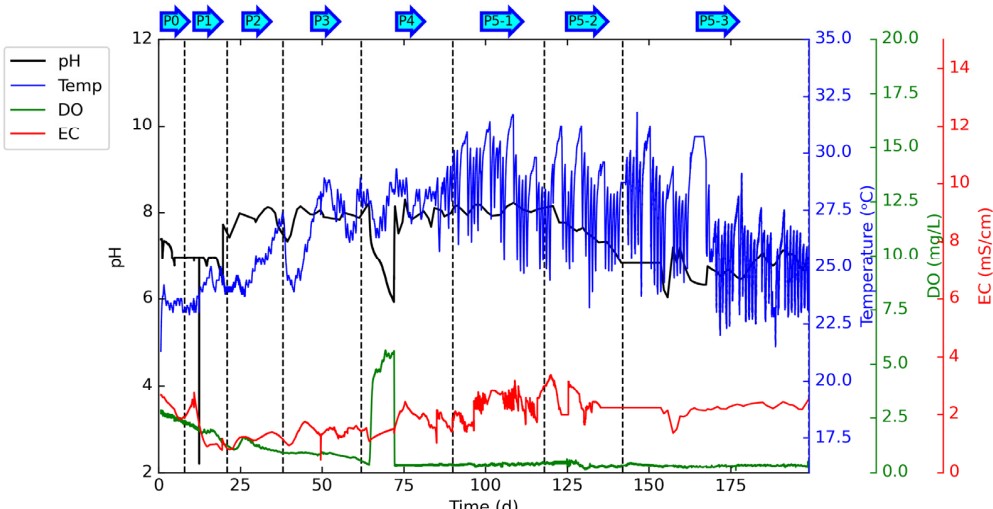

**Figure 1.** The profiles of pH (black line), temperature (blue line), DO (green line), and EC (red line) of the MABR at all phases (P0 to P5-3). Vertical dashed lines were plotted to distinguish different experimental phases. During day 65–72, the influent was ceased while the aeration was kept at a low level (~2 kPa) due to a travel plan, and a boost of DO and a drop of pH was observed.

### 2.3. Water Quality Analysis

During the experiment period, the water quality was measured for influent and effluent aliquots. TOC was measured by a TOC-LCPH analyzer (Shimazu) with an automatic sampler. $NO_2$-N and $NO_3$-N were measured by a Dionex-ICS 5000+ ion chromatography system (Thermo Fisher Scientific) equipped with an AS11-HC column for anion exchange. The following water quality analysis was conducted according to the national standard of the People's Republic of China (with their Standard ID). TN was determined by alkaline persulfate digestion method (HJ 636-2012). $NH_4^+$-N was measured by Nessler's reagent method (HJ 535-2009). The concentration of LAS was determined by methylene blue activated substance (MBAS) method (HJ 826-2017). The chemical oxygen demand (COD) was determined by dichromate method (HJ 828-2017).

### 2.4. Metagenomic Analysis

In total, four groups with 3 samples were taken during P3, P5-1, P5-2, and P5-3 for metagenomic analysis. For each sample, attached biofilm was carefully taken from the hollow fiber surface and then stored in vials at −80 °C before extraction. The DNA was extracted by CTAB method (detailed procedures can be found in the Supplementary Materials), and the purity of extracted DNA was checked by agarose gel electrophoresis. The DNA library was constructed by NEBNext Ultra DNA Library Prep Kit for Illumina. The sequencing was carried out on Illumina PE150 platform.

Raw sequencing data was trimmed by kneaddata (version 0.7.4). Low quality reads were filtered by Trimmomatic (version 0.39) tool and decontamination was performed with bowtie2 (version 2.3.5.1), and data quality was checked with FastQC (version 0.11.9). Taxonomic classification analysis was carried out with Kraken2 (version 2.0.7-beta) and Bracken (version 2.0). The microbial community diversity was analyzed by non-metric multi-dimensional scaling (NMDS). Humann2 (version 2.8.1) was used for functional gene analysis, and the results were compared with protein database UniRef90 with diamond (version 0.8.22) tool. The UniRef90 IDs were referred to Kyoto Encyclopedia of Genes and Genomes (KEGG) database to plot the pathways. The visualization of the metagenomic analysis was conducted on an online platform (www.bioincloud.tech, accessed on 1 November 2022) with R software package provided by Wekemo Tech Group Co., Ltd. (Shenzhen, China).

## 3. Results and Discussion

### 3.1. Performance of Membran-Aerated Biofilm Reactor

3.1.1. Rate and Efficiency for Nitrogen Conversion

To analyse the nitrogen conversion with the MABR, besides TN concentration of the influent and effluent, $NH_4^+$-N, $NO_2$-N, and $NO_3$-N for effluent samples were also measured. Figure 2A illustrates the profiles influent TN, effluent TN, $NH_4^+$-N, $NO_2^-$-N, and $NO_3^-$-N concentration, and the efficiency for ammonia oxidation (based on remnant $NH_4^+$-N and influent TN) and denitrification (based on effluent TN and influent TN). Since the reactor had both oxic and anoxic environment, simultaneous nitrification and denitrification (SND) was expected to take place. It was clear that the influent TN concentration was progressively rising as the feed strength was increasing until full strength EPB wastewater was used. From P1 to P5-3, the influent TN concentration was $112.2 \pm 13.7$, $163.0 \pm 16.5$, $271.6 \pm 28.6$, $450.3 \pm 18.8$, $522.8 \pm 29.5$, $602.5 \pm 33.4$, and $555.6 \pm 39.2$ mg/L, respectively; the effluent TN concentration was $64.3 \pm 13.5$, $121.8 \pm 11.4$, $129.7 \pm 17.2$, $155.8 \pm 17.3$, $216.1 \pm 22.7$, $233.2 \pm 53.7$, and $94.9 \pm 14.5$ mg/L, respectively. The average TN removal efficiency was 42.7%, 25.3%, 52.2%, 65.4%, 58.7%, 61.3%, and 82.9% for each phase (Figure 2B), and stable denitrification had been observed since P3. Meanwhile, removal of $NH_4^+$-N would benefit the downstream water polishing system; therefore, ammonia oxidation efficiency was also evaluated. The effluent $NH_4^+$-N concentration for each phase was $36.1 \pm 11.7$, $77.5 \pm 9.4$, $104.9 \pm 8.9$, $126.7 \pm 14.9$, $188.2 \pm 22.6$, $176.8 \pm 50.1$, and $39.8 \pm 8.3$ mg/L, respectively. The average ammonia oxidation efficiency was 67.8%, 52.4%, 61.4%, 71.9%, 64.0%, 70.7%, and 92.8%, respectively (Figure 2B). $NO_2^-$N emerged at P1-P3, with average concentration of $17.5 \pm 9.3$, $40.1 \pm 4.7$, and $21.5 \pm 13.9$ mg/L, respectively; then, the $NO_2^-$-N level was below 10 mg/L (P4-P5-3). In contrast, the $NO_3^-$-N concentration was below 10 mg/L from P1-P3, then elevated at P4-P5-3, with average concentration of $21.7 \pm 7.2$, $22.1 \pm 8.3$, $43.4 \pm 5.1$, and $44.3 \pm 10.6$ mg/L for each phase. Nevertheless, $NH_4^+$-N was the dominant species in effluent except for P5-3. The denitrification efficiency was somewhat lower than ammonia oxidation efficiency at early P5-2 (Day 117-127). After cleaning the clog in one submerged pump to regenerate the circulation in the reactor tank, the denitrification efficiency gradually went up to the level of ammonia oxidation.

The profiles of nitrogen pollutant were used to calculate the average concentration and conversion rate for each phase. As illustrated in Figure 2C, since P4, the rate for ammonia oxidation and denitrification reached a plateau. The highest ammonia oxidation rate was about 110 mg N $L^{-1}$ $d^{-1}$ and the highest denitrification rate was about 100 mg N $L^{-1}$ $d^{-1}$. $NH_4^+$-N was the predominant nitrogen species in the effluent, indicating ammonia oxidation was the rate-limiting step of nitrogen metabolism. As the surface area of the aeration membrane was fixed and the aeration pressure was not changed, the oxygen supply rate was probably the limiting factor. On the other hand, denitrification was adequately fast to reduce the $NO_2^-$-N and $NO_3^-$-N, so basically there was no accumulation of $NO_x^-$-N. From P5-1 to P5-3, the nitrogen loading was reduced and effluent $NH_4^+$-N concentration gradually declined. Therefore, a longer HRT should be applied to achieve optimized nitrogen removal. Suppose that an astronaut had about 1.2–1.5 L urine per day and the nitrogen content in urine was 8000 mg/L, then a reactor of current configuration with volume of 96–120 L would be required to achieve nitrogen removal for one astronaut. Improvement on the configuration of the reactor would help compact the system size.

Given that the nitrogen conversion rate was about 100 mg N $L^{-1}$ $d^{-1}$, the reactor could treat about 2.2 g N per day. Meanwhile, the net air flow rate through the module, 400 mL/min, indicating the $O_2$ flow rate through the module was $0.4 \times 1.29 \times 21\% \times 1440 = 156$ g/d (assuming air density 1.29 g/L). If this amount of oxygen was fully flowing through the hollow fibers and utilized by the biofilm, the nitrogen conversion should be significantly enhanced. Therefore, most of the air probably directly diffused to the bulk liquid.

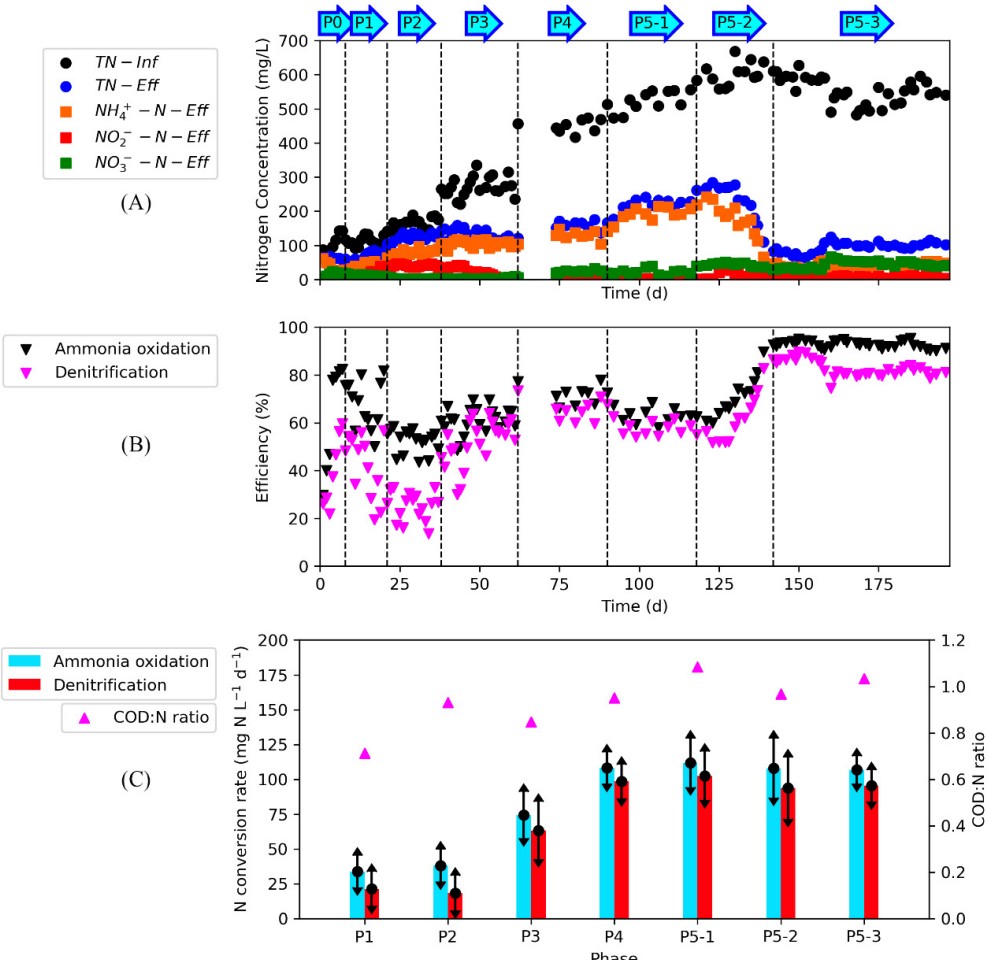

**Figure 2.** The profiles of influent TN concentration (black spots), effluent TN concentration (blue spots), $NH_4^+$-N concentration (orange squares), effluent $NO_2^-$-N concentration (red squares), and $NO_3^-$-N concentration (green squares) from P0 to P5-3 (**A**); the efficiency of ammonia oxidation (black triangles) and denitrification (magenta triangles) of the MABR from P0 to P5-3 (**B**); and the average ammonia oxidation rate (cyan bars), denitrification rate (red bars), and the COD:N ratio (magenta triangles) of the MABR from P1 to P5-3 (**C**). No aliqout was taken during day 65-72 due to a travel.

### 3.1.2. Rate and Efficiency for Pollutant Degradation

The TOC concentration of influent and effluent aliquots was measured to determine the TOC removal efficiency of the MABR system. The results of TOC concentration and the corresponding removal efficiency are illustrated in Figure 3A. The influent TOC concentration was progressively rising as the strength of waste stream increased. From P1 to P5-3, the average influent TOC concentration was $53.4 \pm 13.8$, $91.8 \pm 16.4$, $157.1 \pm 44.7$, $271.8 \pm 54.0$, $373.2 \pm 55.4$, $391.1 \pm 17.8$, and $370.8 \pm 31.1$ mg/L, respectively; the effluent TOC concentration was $3.5 \pm 1.0$, $4.7 \pm 1.2$, $8.2 \pm 2.4$, $20.3 \pm 11.0$, $40.8 \pm 18.0$, $27.3 \pm 20.2$, and $12.2 \pm 4.9$ mg/L, respectively. For most of the time, the TOC removal efficiency was above 90% except P4 and P5-1, which could be attributed to the increase of TOC load, and after switching to a longer HRT (P5-2 and P5-3), the TOC removal efficiency went back above 90%.

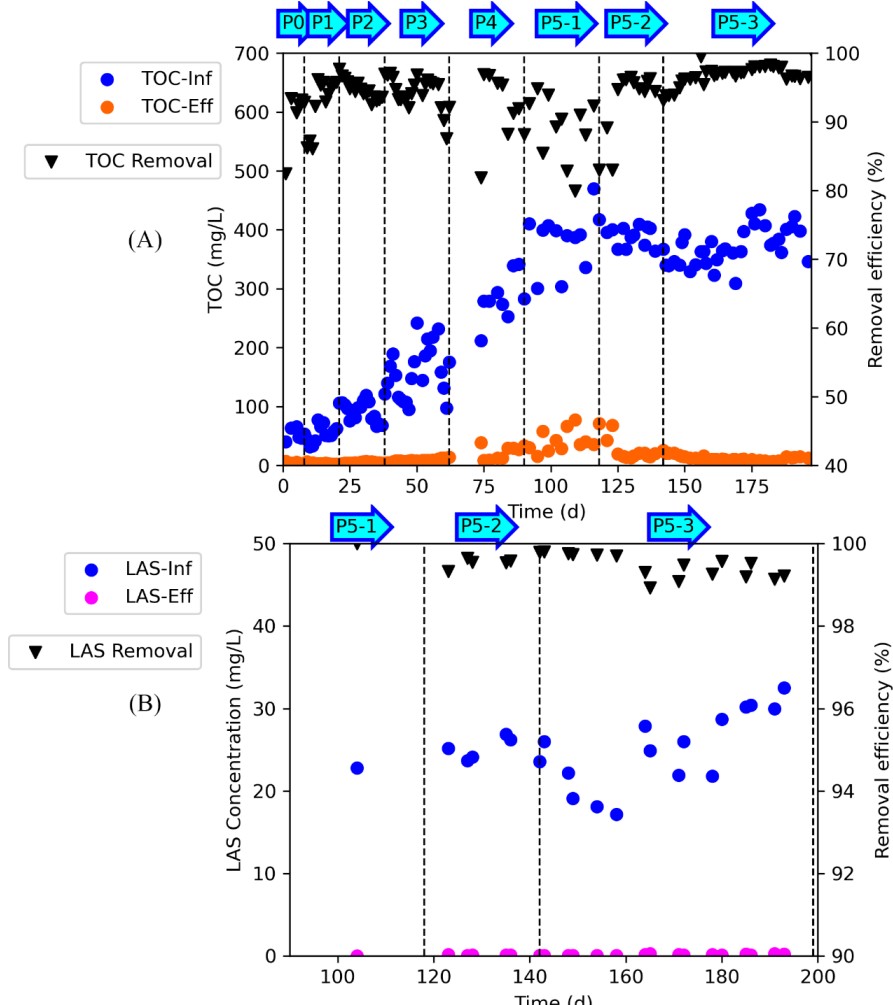

**Figure 3.** The profiles of influent TOC concentration (blue spots), effluent TOC concentration (orange spots), and TOC removal efficiency (black triangles) of the MABR from P0 to P5-3 (**A**); The profiles of influent LAS concentration (blue spots), effluent LAS concentration (magenta spots), and LAS removal efficiency (black triangles) of the MABR at phases P5-1 to P5-3 (full strength EPB wastewater) (**B**).

Since the effluent could potentially be added to the plant hydroponic systems, and according to the National Standards of People's Republic of China (GB 5084-2021), the concentration of anionic surfactants in irrigation water should be lower than 5 mg/L, measurement of LAS concentration was conducted from P5-1 to P5-3 (full strength waste stream was fed). As there were oxic regions in the MABR, high degradation of the surfactants was expected. The LAS concentration profiles are shown in Figure 3B. The average influent LAS concentration was $27.0 \pm 5.29$ mg/L, and the average effluent LAS concentration was $0.14 \pm 0.07$ mg/L, indicating the LAS removal efficiency was about 99.5% and the effluent LAS level was meeting the irrigation water quality standard.

Based on the profiles of TOC concentration and HRT of each phase, the average concentration and TOC removal rate for P1 to P5-3 was 22.3, 38.9, 66.5, 84.1, 111.2, 92.3, and 74.3 mg $L^{-1}$ $d^{-1}$, respectively. As the influent strength increased from P1 to P5-1, the corresponding TOC removal rate also increased up to 111.2 mg $L^{-1}$ $d^{-1}$. The HRT for P5-2 and P5-3 was longer than P5-1 and the TOC load was also reduced, so the corresponding TOC removal rate declined. Although the average TOC removal rate for P5-1 was the highest, the removal efficiency was lower than other phases, and the effluent TOC concentration was about 40 mg/L. To get >90% TOC removal efficiency for EPB waste stream, the TOC loading should be below 100 mg $L^{-1}$ $d^{-1}$.

### 3.1.3. COD Consumption and Denitrification Mechanism

The conventional denitrification process consists of ammonia oxidation, nitrification, and denitrification. In theory, 2.86 mg chemical oxygen demand (COD) would be consumed per mg N removed from the effluent. Other denitrification mechanisms, such as partial nitrification-denitrification (PND) and anaerobic ammonia oxidation (anammox), required lower COD. For PND pathway, 1.71 mg COD would be consumed per mg N removed, and for anammox, the pathway was not involved with COD consumption at all. From P1 to P5-3, the average influent COD concentration was $80.0 \pm 9.01$, $151.8 \pm 17.42$, $230.1 \pm 27.85$, $428.6 \pm 45.6$, $567.1 \pm 64.5$, $582.7 \pm 67.0$, and $574.4 \pm 59.9$ mg/L, respectively; the effluent COD concentration was $15.1 \pm 1.73$, $29.3 \pm 3.76$, $40.5 \pm 4.52$, $60.7 \pm 7.08$, $99.5 \pm 10.64$, $106.0 \pm 12.49$, and $90.6 \pm 10.02$ mg/L, respectively; and the average removal efficiency at each stage was 81.1%, 80.7%, 82.4%, 85.8%, 82.3%, 81.8%, and 84.2%. The experimental COD:N ratio was approximately within the range of 0.8–1.1 (Figure 2C), which was very low compared to typical municipal wastewater. With this amount of COD, the maximal TN removal efficiency by nitrification and denitrification mechanism would be 28.0–38.5; the maximal TN removal efficiency by PND mechanism would be 46.7–64.2%; and the maximal TN removal efficiency by anammox would be 100%. During P3 to P5-2, the TN removal efficiency of MABR was about 60%, and PND could be the major denitrification metabolism. In P5-3, the TN removal efficiency rose to about 80%, so anammox was very likely to participate in the nitrogen metabolism. More details would be provided by molecular biological analysis.

### 3.2. Microbial Community Analysis

The variation of bacterial community structure for biofilm samples taken from P3, P5-1, P5-2, and P5-3 is shown in Figure 4. Proteobacteria was the phylum with the highest microbial abundance (Figure 4A), and the top 20 genera in abundance took 50–60% of the OTUs (Figure 4B). The Venn diagram exhibited the composition of 3459 OTUs for 15 regions (Figure 4C), and the number of OTUs in four different phases was 2919 (P3), 2756 (P5-1), 2832 (P5-2), and 2597 (P5-3), respectively. The microbial community for all samples were highly related, since 2195 OTUs were common for the four groups (75.2–84.5% overlap). Meanwhile, less than 300 OTUs were unique for each group of samples at different phases. The NMDS analysis exhibited that the distance between different groups was close (Figure S2). These results indicated that the microbial community structure of the biofilm in the MABR was stable for consistent nitrogen removal with high efficiency.

The majority of the reads within the microbial community at the phylum level could be attributed to proteobacteria. Proteobacteria has a high level of metabolic diversity of bacteria, which is usually associated with the global carbon or nitrogen cycle and dominates the wastewater treatment process [21]. At the genus level, various genera involved with nitrification and denitrification were found. The presence of ammonia-oxidizing bacteria (AOB), nitrite-oxidizing bacteria (NOB), denitrifying bacteria, and anammox bacteria could provide proof for the high nitrogen removal efficiency of the MABR at the cellular level. Interestingly, the bacteria *Ideonella Sakaiensis* were present in all samples. Since this bacteria utilizes the PET plastic as the sole carbon source [22], and the reactor system (having PMMA tank, PVDF hollow fiber membranes) was free from PET, the PET source was possibly introduced via PET containers during transportation of waste stream.

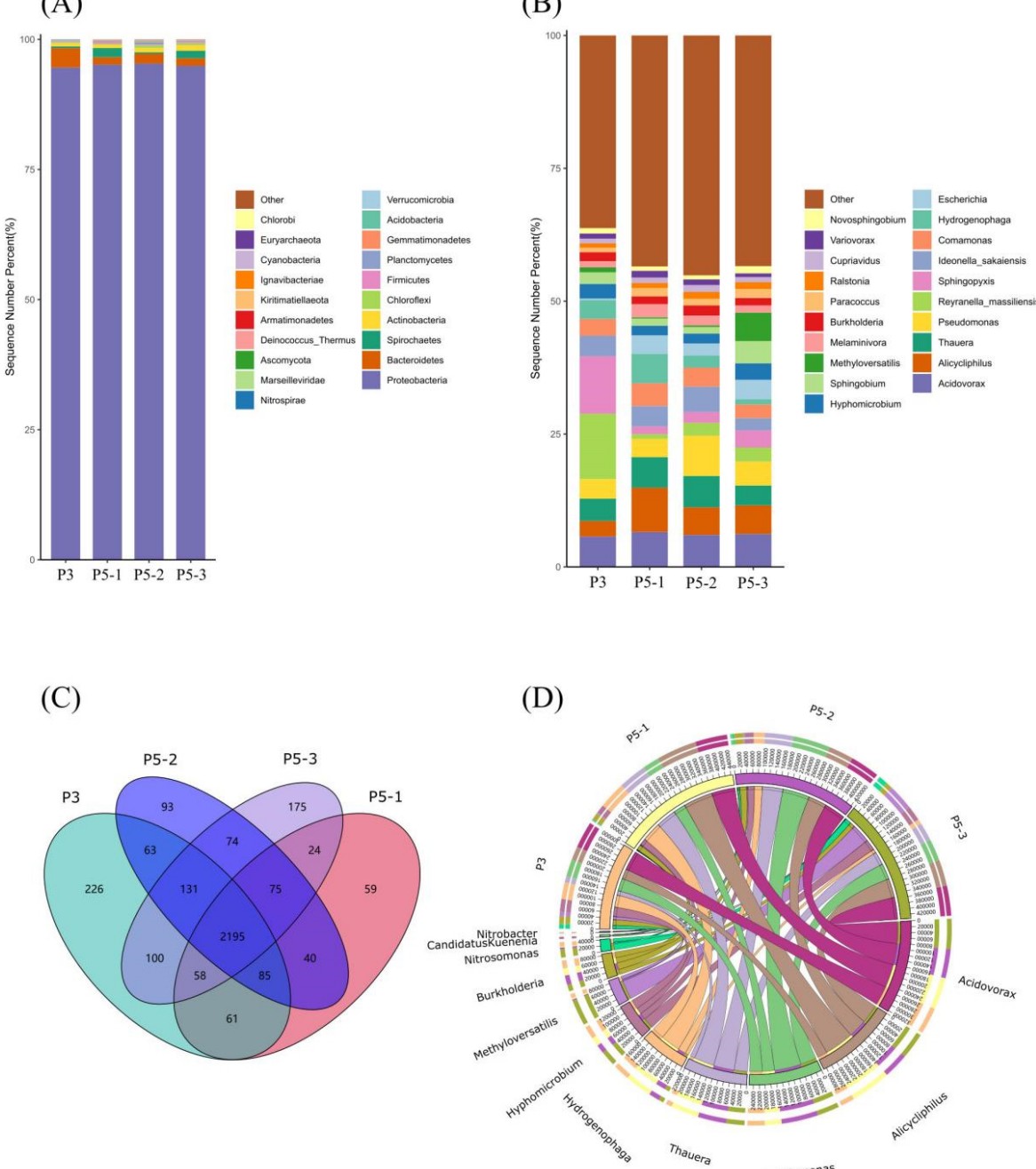

**Figure 4.** Taxonomic classifications of bacterial communities of biofilm samples from the MABR at P3, P5-1, P5-2, and P5-3 at phylum level (**A**) and genus level (**B**). Venn diagram (**C**) exhibits the species composition across the four samples. Circos diagram of prominent functional bacteria for nitrification and denitrification detected in the samples of all groups (**D**). For clarity, categories with abundance lower than top 20 were clustered as "other" in (**A**,**B**).

The nitrifying bacteria and denitrifying bacteria played a key role in the nitrogen removal of the EPB wastewater. Based on the metagenomic results, eight genera with ammonia-oxidation or nitrite-oxidation activity were present, including four AOB: *Nitrosomonas*, *Nitrosospira*, *Nitrosovibrio*, and *Nitrosococcus* and four NOB: *Nitrobacter*, *Nitrococcus*, *Nitrospira*, and *Nitrolancea*. *Nitrosomonas* was the dominant AOB, whereas *Nitrobacter* was the dominant NOB (Table S2). The reads of the rest AOB and NOB were significantly lower, so most of the nitrification process should be carried out by these genera. An anammox bacteria genus *Candidatus Kuenenia* was present, the abundance of which was lower than

AOB and higher than NOB. On the other hand, denitrifying bacteria was predominant in the biofilm samples with high variance. *Acidovorax*, *Alicycliphilus*, *Pseudomonas*, and *Thauera* were well-known denitrifying bacteria with *nirS* gene [23]. *Hydrogenophaga* [21], *Hyphomicrobium* [24], *Methyloversatilis* [25], and *Burkholderia* [26] had also been reported to have capability of denitrification. The abundance of these genera at P3 to P5-3 was visualized by a circos diagram (Figure 4D). Since in the configuration of MABR the bulk liquor was anoxic and only on the surface of aeration hollow fibers existed oxic regions, it was reasonable that denitrifying bacteria was much more abundant than oxygen-demanding nitrifying bacteria. The growth of NOB was limited by the low DO and nitrite concentration, so the abundance was lower than AOB.

The structure of microbial community could also illustrate the efficiency, rate, and mechanism for nitrification and denitrification at the cellular level. $NH_4^+$-N was the main form of nitrogen in effluent until late P5-2, and this trend was consistent with the abundance of AOB, which was low at P5-1, then had a drastic increase at P5-2 to P5-3. As the rate for oxygen supply rate was fixed by the aeration membrane surface area and aeration pressure, and the oxygen might be consumed by heterotrophic microorganisms when influent carbon sources were available, the growth of AOB was probably inhibited if influent loading was high and promoted only when influent loading was appropriate. The abundance of NOB was much lower than AOB, indicating the activity of nitrite-oxidation was low. Therefore, for the majority of heterotrophic denitrifying bacteria, PND was likely the pathway of denitrification, which, compared to denitrification from nitrate, would promote the TN removal efficiency on condition that EPB wastewater had low COD:N ratio. Aside from the regular denitrifying bacteria using organic COD sources, the genus *Hydrogenophaga* was detected in the biofilm samples, indicating hydrogenotrophic denitrification might occur [27]. Since no external hydrogen gas was introduced to the MABR during all experimental phases, the hydrogen gas might be a product of anaerobic degradation of the organic component in EPB wastewater. In spite of this, the anaerobic digestion to $H_2$ would not introduce extra COD and alter the theoretical TN removal efficiency for partial nitrification-denitrification pathway. As mentioned above, the theoretical TN removal efficiency for PND was 46.7–64.2% given the experimental COD:N ratio, so autotrophic denitrification must be involved if higher TN removal efficiency was achieved. The presence of anammox bacteria *Candidatus Kuenenia* could account for the high TN removal efficiency at late P5-2 and P5-3, whose abundance was also higher in samples from P5-2 and P5-3. Since anammox bacteria utilized $NO_2^-$-N and $NH_4^+$-N as substrates, it could also compete with NOB and inhibit its growth. Therefore, the results of microbial community structure were in accordance with the results from the chemical analysis of carbon and nitrogen contaminants.

### 3.3. Functional Genes for Nitrogen Metabolism

The analysis of microbial community structure could illustrate the reactor performance from the cellular perspective, and the analysis on functional genes could interpret the molecular mechanism. As the MABR was a nitrogen removal reactor, the abundance of functional genes present in nitrification, denitrification, as well as anammox pathways were listed (Table S3). Figure 5 illustrates the genes involved in these pathways and their abundance in samples from P3 to P5-3. In total, six genes participating in nitrification pathway were detected (*amoA*, *amoB*, *amoC*, *hao*, *nxrA*, and *nxrB*). These genes encoded ammonia monooxygenase (AMO), hydroxylamine oxidoreductase (HAO), and nitrite oxidoreductase (NXR), respectively, which were essential enzymes for ammonia-oxidation through the hydroxylamine intermediate and nitrification process. Moreover, genes involved in the denitrification pathway were detected, including *narG*, *narH*, *narI*, *napA*, *napB*, *nirK*, *nirS*, *norB*, *norC*, and *nosZ*, which encoded nitrate reductases (*narGHI* and *napAB*), nitrite reductases (*nirK* and *nirS*), nitric oxide reductase (*norBC*), and nitrous-oxide reductase (*nosZ*), covering the denitrification pathway [28]. In addition, genes encoding hydrazine synthase (*hzs*) [29] and hydrazine dehydrogenase (*hdh*) [30] were detected, which were the key enzymes of anammox pathways. The presence of anammox genes and anammox bacteria, together

with the high denitrification efficiency, indicated that anammox pathway contributed to the TN removal of the MABR system.

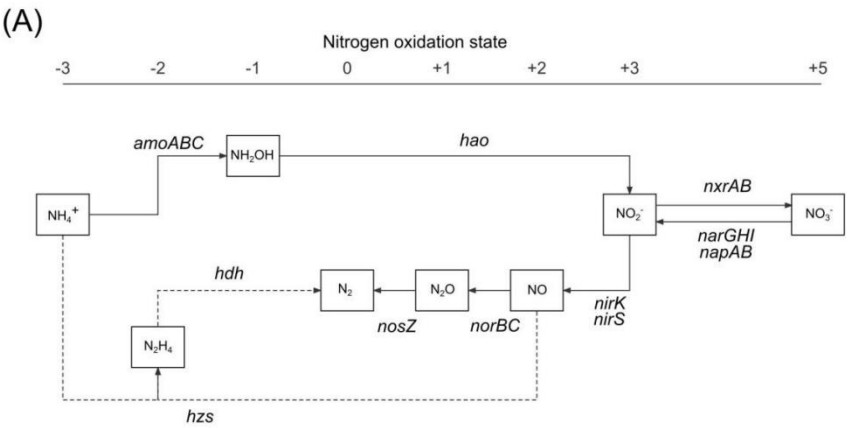

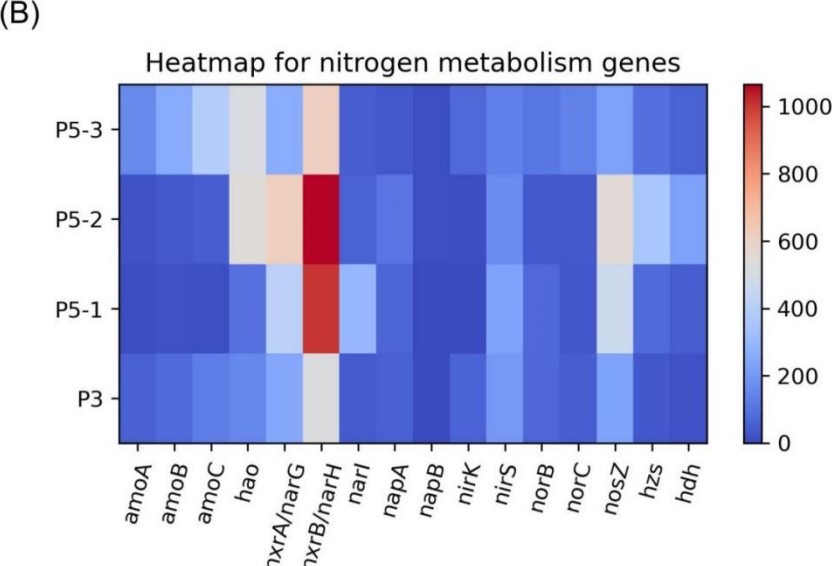

**Figure 5.** Genes involved in nitrification and denitrification pathways (**A**) and the abundance heatmap for functional genes detected in biofilm samples from the MABR at P3, P5-1, P5-2, and P5-3 (**B**).

The gene pairs *nxrA/narG* and *nxrB/narH* had a close genetic relationship and high resemblance [31]; therefore, they were grouped in the same orthologs (KEGG orthology K00370 and K00371). For the rest of the functional genes, the abundance was generally evenly distributed without distinct peaks (Figure 5B). However, the reads for *amoABC* and *hao* had a significant increase at P5-3, and this trend was consistent with the growing trend of AOB observed in microbial community structure. The reads for denitrification genes did not exhibit much fluctuation in any phase, which was in accordance with the stable denitrification rate and efficiency of the MABR. The reads for anammox genes exhibited a peak in P5-2, which matched the peak of anammox bacteria abundance and the rising trend of nitrogen removal efficiency in this phase. Overall, the results of metagenomic analysis on functional genes for nitrogen metabolism could well illustrate the reactor performance and microbial community structure at the molecular level.

## 4. Conclusions

In practice, the MABR unit exhibited 96% removal of TOC and 82% removal of TN when operated under optimized hydraulic retention time for early planetary wastewater

(containing about 10 vol% of urine) with COD:N ratio 0.8–1.1. Metagenomic analysis results indicated that at the cellular level, denitrify bacteria was promoted and anammox bacteria *Candidatus Kuenenia* existed, and at the molecular level, genes involved in nitrification, denitrification, and anammox pathways were present. This demonstration has potential for space applications, including flight and planetary base scenarios, as a technology compatible with reduced gravity conditions. Future improvement in the system would focus on the air tightness of the module, compaction of the reactor volume, and the nitrogen removal rate.

**Supplementary Materials:** The following supporting information can be downloaded at: https://www.mdpi.com/article/10.3390/w14223704/s1, Supplementary material associated with this article can be found in the online version.

**Author Contributions:** Conceptualization, C.Z. and L.Z.; Data curation, C.Z.; Funding acquisition, W.A. and W.D.; Investigation, C.Z.; Methodology, C.Z.; Project administration, W.A.; Supervision, L.Z.; Writing—original draft, C.Z.; Writing—review and editing, C.Z., L.Z. and W.D. All authors have read and agreed to the published version of the manuscript.

**Funding:** This work was financially supported by the National Key Laboratory of Human Factors Engineering, (Grant NO. 6142222200710 and SYFD061908).

**Informed Consent Statement:** Not applicable.

**Data Availability Statement:** The data presented in this study are available on request from the corresponding author.

**Acknowledgments:** The authors sincerely thank Xiangfeng Cao, Yantai Chen, Mengting Hu, Weipeng Yang, and Shanjun Yin for their assistance in water quality tests, and Wekemo Tech Group Co. Ltd. (Shenzhen) for genome sequencing and analysis service.

**Conflicts of Interest:** The authors declare no conflict of interest.

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
