# Peer review of "Green and Sustainable Treatment of Urine Wastewater with a Membrane-Aerated Biofilm Reactor for Space Applications"

_water, doi:10.3390/w14223704_

Round 1

Reviewer 1 Report

In this paper, the authors describe the results of the effectiveness of total organic carbon removal and denitrification in the treatment wastewater, containing urine and hygiene wastewater using the membrane-aerated biofilm reactor developed by the authors. The article is well written, the topic is interesting, and the research is done professionally. This paper is therefore recommended for publication. I wish you good luck in your research work.

Author Response

Thanks for the positive feedback of our manuscript.

Reviewer 2 Report

This is a very interesting study dealing with the treatment and recycle of wastewater in space technology applications. I recommend its publiction after the following minor corrections:

1) The abstract may be shortened for sake of clarity;

2) Can the authors make some comments on the superiority of biological treatment when compared with other promising options (such as photocatalysis and electrocatalysis)?

Reviewer 3 Report

This work investigated the membrane-aerated biofilm reactor for treatment of urine and hygiene wastewater. This is an interesting paper. However, more analyses are needed before this can be accepted for publication. The following issues need clarification:

1.     The Introduction part is lengthy. However, the novelty of this manuscript was not clearly stated.

2.     Page 17, lines 494-496, the manuscript stated that “This demonstration had potential for space applications, including flight and planetary base scenarios, as a gravity-independent technology.” However, there was no any evidence to show that it was a gravity-independent reactor in this manuscript.

3.     Page 4, line 89, why would the biofilm attached on fiber surface be gravity-independent? Please give an explanation.

4.     Page 10, line 306, “the net air flow rate through the module, 400 mL/min” That was a huge amount of air usage. Is the air flow rate of 400 mL/min possible in a real space application?

5.     Page 10, line 309, The statement that “although no bubble was observed during the experiment, it was quite possible there was a leak within the module” is so confusing. First, the leak test was supposed to be conducted before the experiment commenced. Second, if there was a leak within the module, the leak was supposed to be detected and resolved before the experiment began. Third, could the data be trusted if there was a possible leak present?

6.     Page 7, line 203, the statement that “In general, the pH stayed within the range of 7-8” is not rigorous. The pH below 7 was observed for a while during phase P5-3.

7.     Were the hydrophobic PVDF hollow fibers used in this study homemade or commercial? Please specify and provide their basic parameters, such as porosity, thickness, etc.

8.     Use only the levels of heading necessary to differentiate distinct sections in a paper. Please double check if three and four levels of heading were really necessary, e.g., “2.2.1”, “2.2.2”, “2.3.2.1”, “2.3.2.2”, “3.1.1”, “3.1.2”, “3.1.3”, “3.2.1”, “3.2.2”.

9.     Page 6, line 188, please define “EC” “DO” first then use the abbreviation instead.

Typo: There are two “the” in line 207, page 7.

Reviewer 4 Report

The presented work deals with a very interesting, specific and significant problem of ensuring sustainable living conditions for the space crew from the point of view of water reuse and related urine processing. The authors designed and tested a system based on the biochemical oxidation-reduction treatment of urine, which enables the elimination of ammonia nitrogen by a sequence of subsequent processes of nitrification and denitrification, which is commonly used in water treatment and wastewater treatment. I want to highlight the designed sophisticated membrane-aerated biofilm reactor for space applications, which makes it possible to cultivate nitrifying bacteria directly on the surface of aerated iomase carriers and at the same time create conditions for denitrification of oxidized forms of nitrogen in the more distant layers of the biofilm. The validation results of this method revealed the possibility of achieving up to 96% total organic carbon (TOC) removal efficiency and up to 82% denitrification efficiency for an influent with 370–390 mg/l 20 TOC and 500–600 mg/l total nitrogen (TN) without an additional source carbon or sludge discharge. Metagenomic analysis showed the presence of various nitrifying, denitrifying and anammox bacteria in the microbial community and the existence of functional genes in the nitrifying, denitrifying and anammox pathways. Wastewater produced by crew members, including urine and sanitary sewage, can therefore be treated by biological processes to promote recycling efficiency.

Compared to physical and chemical technologies, biological methods are more sustainable, as they generally had lower consumption, milder operating conditions and no hazardous products.

Author Response

Thanks for the positive feedback of the manuscript.

Round 2

Reviewer 3 Report

The advantage of the membrane-aerated biofilm reactor is not equal to the novelty of this manuscript. The novelty of this revised manuscript is still not clear. The response #5 is somewhat arbitrary. Moreover, considering that such high flow rate of air (about 1500 mL/min at inlet) was used and deformation on the acrylic shell was formed under pressure in a long process, it is suspicious that the membrane-aerated biofilm reactor would reduce the cost and promote the recycling efficiency of wastewater treatment.
